# The Effect of Cell-Free Metabolites of Vaginal Lactobacilli on HeLa Cells Is Independent of Lactic Acid Concentration

**DOI:** 10.3390/ijms262411929

**Published:** 2025-12-11

**Authors:** Yulia Myachina, Andrey Sgibnev

**Affiliations:** 1Institute for Cellular and Intracellular Symbiosis of the Ural Branch of the Russian Academy of Sciences, Pionerskaya St. 11, Orenburg 460000, Russia; rogovaya_yulyash@mail.ru; 2Department of Chemistry, Orenburg State Medical University, Sovietskaya St. 6, Orenburg 460000, Russia

**Keywords:** cervical cancer, HeLa cells, *Lactobacillus*, probiotics, oxidative stress

## Abstract

It remains unclear how metabolites produced by vaginal peroxide-producing lactobacilli influence parameters supporting cervical cancer cell survival. The aim of our study was to investigate the functional response of HeLa cells to cell-free metabolites of vaginal lactobacilli producing peroxide under conditions of oxidative stress. HeLa cells were treated with cell-free metabolites of lactobacilli isolated from the vaginal fluid of healthy women. Subsequently, their resistance to oxidative stress (total number of surviving, apoptotic, and necrotic cells), dehydrogenase activity with the MTT assay, and mitochondrial potential were measured. Pretreatment with cell-free lactobacilli metabolites significantly reduced HeLa cell survival under oxidative stress in most cases; dehydrogenase activity and mitochondrial potential changed to a lesser extent. All HeLa cells pretreated with cell-free lactobacillus metabolites that died due to oxidative stress died apoptotic death. These effects of cell-free lactobacilli metabolites are not always determined by lactic acid levels. These data reveal a new mechanism by which vaginal lactobacilli exert local antitumor protection by inducing controlled cell death in transformed cells.

## 1. Introduction

The predominance of lactobacilli (*Lactobacillus* spp.), including H_2_O_2_-producing strains, in the microbial community of the reproductive tract of healthy women is an important factor in protecting vaginal health and preventing urogenital infections [1].

Lactobacilli exert their protective functions through a variety of mechanisms, including competitive exclusion of pathogens, production of antimicrobial compounds (such as lactic acid, bacteriocins, and hydrogen peroxide), and immunomodulation [2,3].

In addition to studies of the role of lactobacilli in infection prevention, there has been growing interest in the direct effects of lactobacilli and their metabolites on host cells and the processes occurring within them, particularly in the context of cancer. For example, results have been described demonstrating the effect of certain lactobacilli strains on a mouse model of colorectal cancer [4]; the potential anticancer effect of lactobacilli mixtures in 3D culture systems has been demonstrated [5]; and L. *crispatus* has been shown to significantly inhibit the growth of cervical cancer cells without significantly affecting normal cervical cells [6].

Cervical cancer, the main cause of which is associated with persistent infection with high-risk human papillomaviruses, remains one of the leading causes of cancer mortality in women worldwide [7]. A characteristic feature of carcinogenesis and tumor progression is a change in the metabolism of transformed cells, which is often accompanied by a disruption in their response to oxidative stress [8]. While moderate levels of reactive oxygen species can stimulate proliferative signals, excessive oxidative stress can trigger cell death, but cancer cells often develop and refine mechanisms to evade this program, which contributes to the development of resistance to therapy [9].

Considering the above, the search for substances capable of modulating the response to oxidative stress and selectively inducing cancer cell death represents a promising direction for the development of adjuvant therapy. Recently, research data has emerged indicating that some probiotic strains and the metabolites they secrete can exert selective antiproliferative and proapoptotic effects on various cancer cell lines, including those of cervical origin [6,10,11,12]. These effects are often associated with the ability of microbial metabolites to disrupt the metabolism and redox homeostasis of cancer cells. However, it remains unclear how metabolites produced by vaginal lactobacilli influence the parameters that support the survival of cervical cancer cells, especially under conditions of oxidative stress.

In this study, we attempted to analyze the functional response of HeLa cervical adenocarcinoma cells to cell-free metabolites of vaginal lactobacilli under induced oxidative stress. To do this, we examined how cell-free metabolites of *Lactobacillus* strains isolated from vaginal fluid samples of healthy fertile women alter the metabolic activity, mitochondrial membrane potential, and cell death pattern of HeLa cells under oxidative stress.

## 2. Results

### 2.1. Effect of Cell-Free Metabolites of Vaginal Lactobacilli on the Survival of HeLa Cells Under Oxidative Stress

Overall, we found that among the lactobacilli selected for the study, there were both isolates that significantly reduced the viability of HeLa cells and isolates that increased the viability of HeLa cells under oxidative stress (Figure 1). The lactobacilli we used in this study produced hydrogen peroxide at concentrations ranging from 0.0 to 2.8 mM when incubated in minimal glucose medium (Appendix A), which is designed to quantify hydrogen peroxide production as described previously [13]. However, in this study, we were unable to assess the effect of hydrogen peroxide produced by these lactobacilli on HeLa cells because the cell-free metabolites we obtained did not contain hydrogen peroxide. This was due to a high rate of hydrogen peroxide degradation in both MRS medium and cell-free supernatants, which was more than 4.8 mM/s (Appendix A), which was higher than the lactobacilli could produce. Co-cultivation of lactobacilli and HeLa cells could ameliorate this situation; however, this was not part of our study design.

We observed that pretreatment of HeLa cells with 5.6 mM lactic acid significantly increased cell survival under oxidative stress (Figure 1, blue column in category k2). In contrast, treatment with 11.1 mM lactic acid had almost no effect on cell survival under oxidative stress (Figure 1, red column in category k2). It is believed that a concentration of 5.6 mM lactic acid is observed in the vagina of women with bacterial vaginosis, lactobacilli deficiency, or their complete absence [3], while 11.1 mM is observed in healthy women with physiological levels of lactobacilli. These facts led us to assume that comparable results could be expected when treating HeLa cells with cell-free lactobacilli metabolites standardized for lactic acid content. However, as it turned out later, the effects of lactic acid were not identical to the effects of cell-free lactobacilli metabolites despite their standardization.

Cell-free metabolites of most lactobacilli (8 out of 10 isolates), in which the concentration of lactic acid was 5.6 mM, were the cause of a decrease to varying degrees in the survival of HeLa cells under oxidative stress conditions (Figure 1, blue columns in categories lb5–lb16 and lb22). We obtained comparable results in the case of treating HeLa cells with cell-free lactobacilli metabolites containing 11.1 mM lactic acid (Figure 1, red columns in categories lb5–lb18).

We were interested in the structure of cell death in HeLa cells. In all cases, HeLa cells died by apoptosis under conditions of oxidative stress after pretreatment with cell-free lactobacilli metabolites, regardless of their lactic acid content (Appendix A). The same effect was observed when HeLa cells were treated with 5.6 mM lactic acid. It should be noted that after pretreatment of HeLa cells with 11.1 mM lactic acid, a small proportion of cells (no more than 5% of the total number of dead cells) died by necrosis, while the rest died by apoptosis.

The presence in our study of strains capable of significantly reducing the survival of HeLa cells under conditions of oxidative stress (isolates lb7, lb9, and lb12), as well as strains that have the opposite effect (for example, isolate lb22), regardless of the content of lactic acid in their cell-free metabolites, indicates the ability of these lactobacilli to produce substances that are still unknown to us, the effect of which we observed.

### 2.2. Effect of Cell-Free Metabolites of Vaginal Lactobacilli Standardized for Lactic Acid Content on the Mitochondrial Potential of HeLa Cells

In this study, we found that oxidative stress induced exclusively apoptotic cell death in HeLa cells pretreated with cell-free lactobacillus metabolites. Given that mitochondria play a key role in activating apoptosis in mammalian cells, we considered it necessary to evaluate how cell-free lactobacillus metabolites influence mitochondrial function and measure mitochondrial potential. To determine changes in the mitochondrial potential of HeLa cells, we used the cationic dye LumiTracker Mito Red CMXRos (Lumiprobe RUS, Moscow, Russia), which accumulates only in metabolically active mitochondria. The degree of its accumulation indicates the level of mitochondrial activity and their integrity.

We found that pre-treatment with lactic acid at a concentration of 5.6 mM did not significantly affect the mitochondrial potential of HeLa cells (Figure 2, blue bar in category k2), whereas 11.1 mM resulted in a nearly two-fold decrease (Figure 2, red bar in category k2). However, cell-free metabolites had unequal effects on the mitochondrial potential of HeLa cells, and this was not determined by the lactic acid content. For example, there was a strain whose metabolites clearly reduced mitochondrial potential. In a few cases, we observed a significant increase in mitochondrial potential (strains lb12 and lb18) or, conversely, a decrease (strains lb5, lb7, lb8, lb13, lb28), and this did not depend on the lactic acid concentration.

At the same time, we did not find a significant correlation between the number of HeLa cells surviving oxidative stress and the changes in their mitochondrial potential after treatment with cell-free lactobacilli metabolites (Spearman’s correlation coefficient r = 0.21, *p* value (two-tailed) = 0.561 and r = 0.26, *p* value (two-tailed) = 0.469 for 5.6 and 11.1 mM, respectively).

Thus, changes in mitochondrial potential did not correlate with either the lactic acid concentration in cell-free metabolites or with changes in the viability of HeLa cells pretreated with cell-free lactobacilli metabolites. These changes were also not associated with the ability of lactobacilli to produce hydrogen peroxide; surfactant content also did not affect mitochondrial potential or HeLa cell viability. This suggests that the lactobacilli metabolites used in our study, in addition to lactate, hydrogen peroxide, and surfactants, may contain other substances capable of influencing mitochondrial potential.

### 2.3. Effect of Cell-Free Metabolites of Vaginal Lactobacilli on the Dehydrogenase Activity of HeLa Cells

In this part of the study, we found an ambiguous effect of lactobacilli cell-free metabolites on the dehydrogenase activity of HeLa cells (Figure 3).

Overall, the degree of inhibition of dehydrogenase activity was not pronounced. Cell-free metabolites of only one strain (lb28), regardless of lactic acid content, significantly increased the level of dehydrogenase activity in HeLa cells; conversely, cell-free metabolites of strains lb5, lb7, and lb9 significantly inhibited dehydrogenase activity. Cell-free metabolites of the remaining strains had insignificant effects on dehydrogenase activity.

Several specific effects of extracellular metabolites are noteworthy. For example, extracellular metabolites of strain lb7 containing 11.1 mM lactate (Figure 3, red column in the lb7 category) reduced dehydrogenase activity to a greater extent than those containing 5.6 mM lactic acid (Figure 3, blue column in the lb7 category).

We observed the same trend for strains lb8, lb12, and lb22, whose cell-free metabolites containing 11.1 mM lactic acid (Figure 3, red columns) had a greater effect than those containing 5.6 mM (Figure 3, blue columns). This difference could be explained by the higher concentration of lactic acid (11.1 vs. 5.6 mM), but we observed cases where the effects of the metabolites were almost equal, regardless of their lactic acid content (Figure 3, categories lb5, lb9, lb16). Moreover, cell-free metabolites of strain lb13 containing 11.1 mM lactic acid (Figure 3, red column in category lb13) reduced dehydrogenase activity to a lesser extent than the same one containing 5.6 mM lactic acid.

Taking all this into account, and the fact that lactic acid in the 11.1 mM control (Figure 3, red column in category k2) itself significantly inhibited the dehydrogenase activity of HeLa cells, we suggest that all these effects of cell-free metabolites may be determined by something other than lactic acid; perhaps these substances can counteract the effects of lactic acid.

At the same time, we found a positive relationship between the inhibition of dehydrogenase activity and the death of HeLa cells under oxidative stress conditions after pretreatment with cell-free metabolites of lactobacilli (Spearman’s correlation coefficient r = 0.77; *p* value (two-sided) = 0.013 and r = 0.87; *p* value (two-sided) = 0.002, for 5.6 and 11.1 mM, respectively).

Thus, the overall conclusion of our study is that lactobacilli metabolites can reduce HeLa cell survival under oxidative stress. This is consistent with the ability of cell-free lactobacilli metabolites (regardless of their lactic acid content) to significantly reduce dehydrogenase activity and, to a lesser extent, the membrane potential of HeLa cells.

## 3. Discussion

In this study, we observed pronounced antiproliferative and proapoptotic effects of cell-free metabolites of vaginal lactobacilli on HeLa cells.

An important finding of our study is that complexes of substances contained in cell-free metabolites of lactobacilli are more effective in suppressing cervical carcinoma cell proliferation than individual substances (such as lactic acid) produced by them. This finding appears to be significant in expanding our understanding of the potential use of lactobacilli, which produce complexes of substances that can inhibit cervical carcinoma cell proliferation, for the development of new cervical cancer treatment regimens. Pretreatment of HeLa cells with low concentrations of lactic acid (5.6 mM) increased their survival under oxidative stress compared to the control. However, after pretreatment with cell-free lactobacilli metabolites containing the same lactic acid concentration, HeLa cell survival under oxidative stress conditions was significantly reduced in most cases.

We observed a more pronounced reduction in HeLa cell survival under oxidative stress after treatment with cell-free metabolites containing a higher (11.1 mM) concentration of lactic acid, although lactic acid itself at this concentration slightly reduced survival. In both cases of cell-free metabolite treatment, regardless of their lactic acid content, we observed lactobacilli strains whose metabolites increased HeLa cell survival under oxidative stress. This allows us to conclude that, although lactic acid concentration may be important in some cases, it is not the only determining factor enhancing carcinoma cell death due to oxidative stress. We believe that the cell-free metabolites of the lactobacilli used in this study contain certain substances we did not account for that were responsible for the observed effects. This, in turn, raises the question of the need to include another property in the criteria for selecting bacterial strains as potential probiotics for vaginal use: the ability to inhibit the proliferation of cancer cells. This seems particularly important given the high prevalence of HPV infection, especially with high-risk oncogenic viruses [14].

Our experiments showed that cell-free metabolites of vaginal lactobacilli caused a significant decrease in dehydrogenase activity, and this, under conditions of oxidative stress, was accompanied by a decrease in the total number of viable cells and the induction of apoptosis, but not necrosis.

In all cases, HeLa cells exposed to oxidative stress died exclusively via the apoptotic pathway, even though the cell-free metabolites of vaginal lactobacilli contained high concentrations of lactic acid, whereas lactic acid itself, at high concentrations, led to cell necrosis.

This suggests that these cell-free metabolites contain other, yet unidentified, sub-stances capable of modulating the cytotoxic effects of lactic acid, causing HeLa cell death under oxidative stress to become apoptotic.

Several mechanisms can be proposed to explain this phenomenon. Firstly, the cell-free metabolites of vaginal lactobacilli may contain biologically active molecules, such as peptides or other low-molecular-weight metabolites other than lactic acid, which can specifically activate caspase cascades, which are key pathways of apoptosis [15]. Secondly, a synergistic effect of several components of cell-free metabolites is likely, which may cause mitochondrial membrane depolarization and cytochrome C release, which is a classic scenario for triggering the intrinsic apoptotic pathway, without the rapid collapse of energy metabolism characteristic of necrosis [15].

Importantly, metabolites from different lactobacilli strains exerted different effects on the metabolic status and cell death patterns of cervical carcinoma cells. This highlights the importance of individual selection of probiotic strains for therapy, as well as the need to study strain-specific effects rather than generalizing our findings to the entire *Lactobacillus* genus.

Thus, the ability of a complex of vaginal lactobacilli metabolites to induce apoptotic cell death in cervical carcinoma cells, rather than necrosis, is of significant inter-est and opens up prospects for the development of new strategies for maintaining vaginal health and adjuvant therapy using probiotics or their metabolites, aimed at reducing the side effects of standard treatment and minimizing inflammatory responses in vivo [16].

Despite the undeniable importance of our data, this study has several limitations. The primary drawback is the lack of a complete characterization of the composition of the cell-free lactobacilli metabolites used in this study. The effect on HeLa cells could be explained by the synergistic action of a whole complex of metabolites (lactic acid, short-chain fatty acids, peptides, etc.), but the contribution of each of these metabolites still needs to be studied in detail. Performing metabolomic analysis (LC-MS/MS) to compare the metabolite profiles of the strains we used in this study is a priority for our ongoing research. Another limitation of this study was that the design of our experiments did not allow us to evaluate the effect of hydrogen peroxide, which was produced by the lactobacilli cultures we isolated, due to the rapid decomposition of hydrogen peroxide. Hydrogen peroxide is of interest in this context, as it is an important factor in antimicrobial and potentially cytotoxic activity [17,18], so the design of subsequent experiments should probably include the co-cultivation of HeLa cells and H_2_O_2_-producing lactobacilli; however, whether this is feasible remains to be seen.

## 4. Materials and Methods

### 4.1. Participants

Thirty female volunteers who met the inclusion and exclusion criteria participated in the study. Inclusion criteria included being 18–30 years of age and providing written, voluntary, informed consent. Exclusion criteria included smoking, immunodeficiency, somatic diseases in the sub- and decompensated stages, menstrual dysfunction, pregnancy, lactation, hormonal contraception, STIs, including HPV infection, and antimicrobial therapy within the month preceding the study. A single vaginal fluid sample was collected from the volunteers on days 7–8 of their menstrual cycle for subsequent isolation of lactobacilli.

### 4.2. Isolation and Identification of Lactobacilli

Lactobacilli from vaginal fluid samples were isolated by plating on MRS agar (HiMedia LPL, Mumbai, India) followed by incubation in an atmosphere with 5% CO_2_ for 48 h. Microorganisms obtained from isolated individual colonies were identified using a combination of morphological, cultural, and biochemical characteristics (API 50 CH test kit according to the manufacturer’s instructions), and additionally by RT-PCR with species-specific primers [19] on a MiniOpticon device (Bio-Rad Laboratories, Inc., Hercules, CA, USA). The ability of isolated lactobacilli to produce hydrogen peroxide and surfactants was determined as described previously [13]. A total of thirty-two lactobacilli isolates were isolated; each assigned a unique number. The total number of *Lactobacillus* species detected, and their characteristics are shown in Appendix A. Ten isolates were subsequently selected for further experiments using an online random number generator https://www.random.org/integers/ (accessed on 9 June 2025).

### 4.3. Obtaining Cell-Free Metabolites of Lactobacilli

To obtain cell-free metabolites, previously isolated lactobacilli were grown in MRS broth at 37 °C in a 5% CO_2_ atmosphere for 24 h (to approximately 10^9^ CFU/mL). The bacterial cultures were centrifuged at 3000× *g* for 20 min. The supernatants were filter-sterilized (0.22 µm) and used as cell-free metabolites. To determine the lactic acid concentration in CFM, the D-Lactate Assay Kit and L-Lactate Assay Kit (Sigma Aldrich Co., St. Louis, MI, USA) were used according to the manufacturer’s instructions.

### 4.4. HeLa Cell Cultivation

We used the HeLa V human carcinoma cell line purchased from Biolot (Saint Petersburg, Russia). Cells were grown at 5% CO_2_ at 37 °C in a complete growth medium consisting of DMEM (Dulbecco’s Modified Eagle’s Medium, Biolot) with high glucose (25 mM), no pyruvate, and supplemented with gentamicin sulfate (50 μg/mL) and 10% fetal bovine serum (Biolot, Saint Petersburg, Russia) as described previously [20].

### 4.5. Effect of Lactobacillus Cell-Free Metabolites on HeLa Cells

Before the study, cells were removed from the surface of the culture flask using a trypsin-versene mixture (Biolot, Saint Petersburg, Russia) and counted in a hemocytometer.

To determine the effect of cell-free metabolites on HeLa cells, 50 µL of CFM, pre-standardized for lactic acid content (to a final lactic acid concentration of 5.6 and 11.1 mM), were added to 450 µL of a cell suspension in DMEM (10^5^ cells/mL) and incubated for 1 h at 37 °C in 5% CO_2_. MRS broth and a lactic acid solution in MRS broth at concentrations of 5.6 and 11.1 mM were used as controls. The total number of viable cells, the number of apoptotic and necrotic cells, dehydrogenase activity, mitochondrial potential, and oxidative stress resistance were then measured.

The total number of viable HeLa cells was measured by staining with 0.4% trypan blue (Sigma Aldrich Co., St. Louis, MI, USA) at a 1:1 dye-to-cell suspension ratio, followed by light microscopy. Viable cells remained colorless, while dead cells were stained blue [21].

The Annexin V-AF488 kit (Lumiprobe RUS, Moscow, Russia) was used to measure apoptotic and necrotic cell counts according to the manufacturer’s instructions.

To determine changes in mitochondrial potential, HeLa cells were stained with the cationic dye LumiTracker Mito Red CMXRos (Lumiprobe RUS, Moscow, Russia) according to the manufacturer’s instructions.

To measure the dehydrogenase activity of HeLa cells, the MTT (3-[4,5-dimethylthiazol-2-yl]-2,5 diphenyl tetrazolium bromide) (Lumiprobe RUS, Moscow, Russia) assay was used [22]. Briefly, MTT was added to a HeLa cell suspension and incubated for 4 h at 37 °C in the presence of oxygen. After pelleting the cells by centrifugation (3000× *g*, 10 min), the supernatant was collected, and dimethyl sulfoxide (DMSO) (Sigma Aldrich Co., St. Louis, MI, USA) was added to the pellet to dissolve the formazan. The optical density of the extracted formazan was measured at 520 nm.

Oxidative stress was induced in HeLa cells as described previously [23]. To measure the effect of cell-free metabolites on the resistance of HeLa cells to oxidative stress, 30 μL of 3% H_2_O_2_ solution was added to a cell suspension (10^5^ cells/mL, 200 μL) pre-treated as described above with cell-free metabolites standardized for lactic acid content three times at hourly intervals, and the cells were then incubated for 24 h at 5% CO_2_, 37 °C. The total number of living, apoptotic, and necrotic cells was then measured.

### 4.6. Statistical Analysis Methods

Experimental data are presented as the mean values of at least three independent experiments with standard deviation (mean ± SD). The Mann–Whitney U test was used to assess differences between mean values. A two-way analysis of variance (ANOVA) was used to assess the significance of differences in the effects of lactobacilli of cell-free metabolites standardized to lactate. The relationship between the survival of Hela cells under oxidative stress, their dehydrogenase activity and mitochondrial potential was determined using Spearman correlation analysis. The threshold of statistical significance was set as a two-tailed *p* value = 0.05.

## 5. Conclusions

We found that cell-free metabolites of lactobacilli isolated from healthy, fertile women reduced the survival of HeLa cells under oxidative stress; HeLa cells died only by apoptosis. A significant positive correlation was established between the inhibition of dehydrogenase activity and the death of HeLa cells under oxidative stress after pre-treatment with cell-free metabolites of lactobacilli. These effects of cell-free metabolites of lactobacilli are not determined solely by lactic acid levels and may depend on other strain characteristics. This underscores the need for individual selection of probiotic strains for therapy, as well as the need to study strain-specific effects rather than generalizing our results to the entire *Lactobacillus* genus.

## Figures and Tables

**Figure 1 ijms-26-11929-f001:**
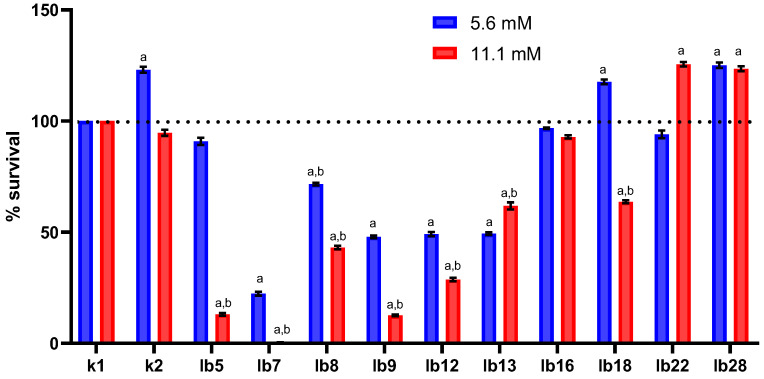
Effect of cell-free metabolites of vaginal lactobacilli standardized for lactic acid content on the survival of HeLa cells under oxidative stress. Horizontal axis legend: k1—cells treated with MRS; k2—cells treated with lactic acid solution in MRS (5.6 or 11.1 mM); lb5–lb28—cells treated with cell-free lactobacilli metabolites pre-standardized for lactic acid content (5.6 or 11.1 mM). Note: Measurements were taken 1 h after treatment. Labels above the columns: a—the data differ significantly from the control (HeLa cells treated with MRS); b—differences between the effects of metabolites containing 5.6 and 11.1 mM lactic acid are significant.

**Figure 2 ijms-26-11929-f002:**
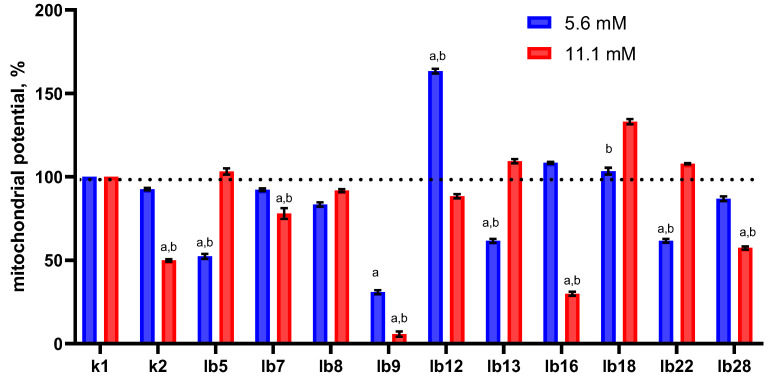
Effect of cell-free metabolites of vaginal lactobacilli standardized for lactic acid content on the mitochondrial potential of HeLa cells. Horizontal axis legend: k1—treated with MRS; k2—cells treated with lactic acid solution in MRS (5.6 or 11.1 mM); lb5–lb28—cells treated with cell-free lactobacilli metabolites pre-standardized for lactic acid content (5.6 or 11.1 mM). Note: Measurements were taken 1 h after treatment. Labels above the columns: a—the data differ significantly from the control (HeLa cells treated with MRS); b—differences between the effects of metabolites containing 5.6 and 11.1 mM lactic acid are significant.

**Figure 3 ijms-26-11929-f003:**
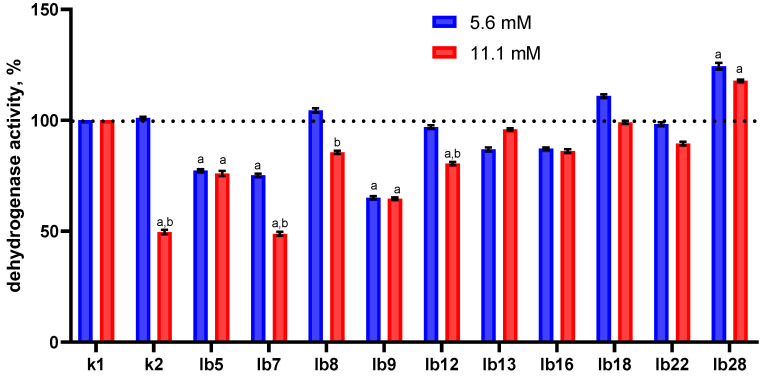
Effect of cell-free metabolites of vaginal lactobacilli standardized for lactic acid content on the dehydrogenase activity of HeLa cells. Horizontal axis legend: k1—cells treated with MRS; k2—cells treated with lactic acid solution in MRS (5.6 or 11.1 mM); lb5–lb28—cells treated with cell-free lactobacilli metabolites pre-standardized for lactic acid content (5.6 or 11.1 mM). Note: Measurements were taken 1 h after treatment. Labels above the columns: a—the data differ significantly from the control (HeLa cells treated with MRS); b—differences between the effects of metabolites containing 5.6 and 11.1 mM lactic acid are significant.

## Data Availability

The data presented in this study are available upon request from the corresponding author due to the need to obtain additional permissions to transfer confidential information related to the volunteers participating in this study.

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
