# Peer review of "The Effect of Cell-Free Metabolites of Vaginal Lactobacilli on HeLa Cells Is Independent of Lactic Acid Concentration"

_ijms, 2025, doi:10.3390/ijms262411929_

Round 1

Reviewer 1 Report

Comments and Suggestions for Authors

The article is devoted to an important and relevant topic, but there are a number of questions about the presented research:

The article says: Lactobacillus metabolites used in our study, in addition to lactate, contain other substances that can influence mitochondrial potential. What exactly can these metabolites be?

How can we justify the choice of dehydrogenase activity as a criterion of survival?

If I understand correctly, the researchers identified only D and L lactate as metabolites of the vital activity of various lactobacilli, whether other metabolites were somehow taken into account

The S1 supplemental materials file contains the names of the lactobacillus strains that have been identified. Some names are repeated several times. What is the reason for this? How was this taken into account?

Author Response

Dear reviewer, thank you very much for taking the time to review this manuscript. Below you will find detailed responses, as well as relevant corrections highlighted in the resubmitted files.

Comments 1: The article says: Lactobacillus metabolites used in our study, in addition to lactate, contain other substances that can influence mitochondrial potential. What exactly can these metabolites be?

Response 1:

Dear reviewer, we appreciate this comment. In addition to lactate, we measured the production of hydrogen peroxide and surfactants in lactobacilli (details are provided in Spreadsheet S1). We found that, although most lactobacilli were capable of producing hydrogen peroxide, no hydrogen peroxide was detected in the metabolites used in the experiment. We attribute this to the rapid decomposition of hydrogen peroxide in both MRS and cell-free metabolites, and we support this with additional experiments demonstrating that hydrogen peroxide is indeed rapidly decomposed. The results of these experiments are presented in Spreadsheet S2. We also found no correlation between the ability of lactobacilli to produce surfactants and their effect on the mitochondrial potential of HeLa cells. We acknowledge that we formulated our idea incorrectly, and we have attempted to correct this in the text of the manuscript as follows: Thus, changes in mitochondrial potential did not correspond either to the concentration of lactic acid in cell-free metabolites or changes in the viability of HeLa cells pretreated with cell-free lactobacilli metabolites. These changes were also not determined by the ability of lactobacilli to produce hydrogen peroxide; surfactant content also did not affect mitochondrial potential or the viability of HeLa cells. This suggests that the lactobacilli metabolites used in our study, in addition to lactate, hydrogen peroxide, and surfactants, may contain other substances capable of influencing mitochondrial potential (lines 152–158, highlighted in yellow). Additionally, in the Discussion section, when discussing the limitations and shortcomings of the study, we added that "Performing metabolomic analysis (LC-MS/MS) to compare the metabolite profiles of the strains we used in this study is a priority for our ongoing research" (lines 265–267, highlighted in yellow).

Comments 2: How can we justify the choice of dehydrogenase activity as a criterion of survival?

Response 2: Dear reviewer, thank you for your comment. In subsection 4.5 (Effect of Lactobacillus cell-free metabolites on HeLa cells), it is noted that we determined the total number of viable HeLa cells using 0.4% trypan blue staining. We used dehydrogenase activity primarily as a criterion for assessing the metabolic activity of HeLa cells. We agree that dehydrogenase activity can be used as an additional criterion for assessing viability, since the MTT assay measures the activity of NAD(P)H-dependent dehydrogenases. These enzymes are active only in living, metabolically competent cells. A decrease in their activity directly correlates with loss of viability, induction of apoptosis, or disruption of metabolic pathways. In addition, we confirm the results of the MTT assay by assessing the number of viable cells using trypan blue.

Comments 3: If I understand correctly, the researchers identified only D and L lactate as metabolites of the vital activity of various lactobacilli, whether other metabolites were somehow taken into account

Response 3: Dear reviewer, we appreciate your comment. As noted above, in addition to lactate, we measured hydrogen peroxide and surfactant production (this is indicated in Spreadsheet S1). We found that, although most lactobacilli were capable of producing hydrogen peroxide, no hydrogen peroxide was detected in the metabolites used in the experiment. Apparently because of this, we found no correlation between the ability of lactobacilli to produce H2O2 and their effect on HeLa cells. We also found no correlation between the ability of lactobacilli to produce surfactants and their effect on the mitochondrial potential of HeLa cells.

At this stage of the study, we specifically standardized and controlled the concentration of lactic acid, the main and quantitatively predictable lactobacilli metabolite, in MRS. This was necessary for a valid comparison of different strains. A complete identification of the entire spectrum of metabolites was not the direct goal of this study.

Comments 4: The S1 supplemental materials file contains the names of the lactobacillus strains that have been identified. Some names are repeated several times. What is the reason for this? How was this taken into account?

Response 4: Dear reviewer, thank you for this comment. Indeed, Table S1 contains strains belonging to a single species. This can be explained by the fact that the number of lactobacilli species in the vagina is limited. Some researchers attribute this to the specific habitat conditions of the vagina. The lactobacilli isolates we used in the study, all belonging to the same species, were obtained from different volunteers; therefore, they are of different origins and are not clones of a single isolate. Moreover, they differ in the levels of lactate, H2O2, and surfactant production. In two cases (volunteer 9 and volunteer 23, Table S1), we were able to isolate two lactobacilli strains; these strains belonged to different species.

Reviewer 2 Report

Comments and Suggestions for Authors

Anything which can be shown to trigger the death of tumour cells is a worthwhile study.  This investigation looked at the culture supernatants of bacteria isolated from the vaginas of healthy volunteers.  My main reservation is that the bacteria were grown in MRS media and inevitably will produce organic acid as a result of their particular metabolism.  Some tumour cells are acid sensitive and not much information was provided about the chemical nature of the cell free metabolites used to treat the HeLa cells.  This is key and at least metabolite characteristics such as pH were needed.  I was surprised that mass spectroscopy analysis of the cell free metabolites was not included as this could have provided insight into what was triggering the HeLa apoptosis.

Also, were the lactic acid bacteria used to produce the cell free metabolites from the current study as in the methods it states the bacteria were previously isolated?

Author Response

Dear reviewer, thank you very much for taking the time to review this manuscript. Below you will find detailed responses, as well as relevant corrections highlighted in the resubmitted files.

Comments 1: Anything which can be shown to trigger the death of tumour cells is a worthwhile study. This investigation looked at the culture supernatants of bacteria isolated from the vaginas of healthy volunteers. My main reservation is that the bacteria were grown in MRS media and inevitably will produce organic acid as a result of their particular metabolism. Some tumour cells are acid sensitive and not much information was provided about the chemical nature of the cell free metabolites used to treat the HeLa cells. This is key and at least metabolite characteristics such as pH were needed. I was surprised that mass spectroscopy analysis of the cell free metabolites was not included as this could have provided insight into what was triggering the HeLa apoptosis.

Response 1:

Dear reviewer, thank you for your comment. We fully agree with you that the specific metabolism of lactobacilli in MRS medium and the production of organic acids (primarily lactic acid) are critical factors that needed to be controlled. This is why we determined the lactate content of cell-free metabolites (details are provided in Spreadsheet S1); furthermore, our experimental design initially included appropriate MRS controls with the addition of lactate. In addition to lactate, we determined the ability of lactobacilli to produce H2O2 and surfactants. We note in the manuscript that neither the ability of lactobacilli to produce H2O2 nor surfactants had a noticeable effect on HeLa cells. To mitigate the nonspecific effects of acidification, all cell-free metabolites were standardized for lactic acid content (5.6 and 11.1 mM) before experiments with HeLa cells. These lactate concentrations are described as those typically found in vaginal fluid. We believe this allowed us to compare the activity of different strains under equal conditions.

Sterile MRS medium, including supplemented with the same lactate concentrations as the cell-free lactobacilli metabolites, was used as one of the controls in all experiments. This allows us to directly assess the contribution of the MRS medium itself and lactate to the observed effect.

The results presented in this article demonstrate that the cytotoxic effect of the most active bacterial supernatants was significantly higher than that of the control acidified MRS medium. This clearly indicates that HeLa cell apoptosis is induced not only by low pH/lactic acid, but also by other biologically active metabolites specifically produced by the strains under study.

We thank you for this crucial recommendation to use mass spectroscopic analysis. You are absolutely right that identifying these specific metabolites is the next logical goal. In the original version of the manuscript, we focused on proving the presence of additional (non-acid-dependent) activity. Additionally, in the Discussion section, when discussing the limitations and shortcomings of the study, we added that "Performing metabolomic analysis (LC-MS/MS) to compare the metabolite profiles of the strains we used in this study is a priority for our ongoing research" (lines 265–267, highlighted in yellow).

Comments 2: Also, were the lactic acid bacteria used to produce the cell free metabolites from the current study as in the methods it states the bacteria were previously isolated?

Response 2: Dear reviewer, thank you for your comment. This study utilized lactobacilli strains previously isolated from the vaginas of healthy donors and currently (since this study) deposited in the collection. Their taxonomic affiliation was determined using a combination of morphological, cultural, and biochemical characteristics (API 50 CH test kit according to the manufacturer's instructions), and additionally by RT-PCR with species-specific primers. In addition, the ability of these same lactobacilli to produce lactate, hydrogen peroxide, and surfactants was measured (Spreadsheet S1). Cell-free metabolites of these same lactobacilli were then used for further experiments with HeLa cells.

Round 2

Reviewer 1 Report

Comments and Suggestions for Authors

I believe that the authors have fully answered the questions raised and the manuscript can be published in the IJMS journal.

Reviewer 2 Report

Comments and Suggestions for Authors

Ideally, more information on the lactic acid bacteria metabolites responsible for inducing apoptosis in the HeLa cells would have been helpful.  However, with the additional methods information added to the manuscript combined with an undertaking to carry out the mass spec characterisations I think this paper now has more merit.